# Hwa-Byung (Anger Syndrome) in the MZ Generation of Republic of Korea: A Survey

**DOI:** 10.3390/jcm13226667

**Published:** 2024-11-06

**Authors:** Chan-Young Kwon, Ju Yeob Kim, So Yeong Park

**Affiliations:** 1Department of Oriental Neuropsychiatry, College of Korean Medicine, Dong-Eui University, 52-57 Yangjeong-ro, Busanjin-gu, Busan 47227, Republic of Korea; 2Department of Korean Medicine, College of Korean Medicine, Dong-Eui University, 52-57 Yangjeong-ro, Busanjin-gu, Busan 47227, Republic of Korea; jykim507@naver.com (J.Y.K.); qkrthdudsksh@naver.com (S.Y.P.)

**Keywords:** generation MZ, mental health, hwa-byung, anger syndrome

## Abstract

**Background/Objectives**: Hwa-byung (HB), traditionally observed in middle-aged Korean women, was investigated among the MZ generation in Republic of Korea to investigate its prevalence, associated factors, and perceptions. **Methods**: An online survey was conducted with 449 Korean adults in the Republic of Korea born between 1980 and 2005. The participants completed questionnaires that assessed HB symptoms, emotional labor, psychological distress, and HB perceptions. Data were analyzed using Chi-squared tests, *t*-tests, and bivariate logistic regression. **Results**: The prevalence of HB in the MZ generation was 36.3%. Significant HB predictors included female sex (odds ratio [OR] = 2.314), poor subjective health (OR = 3.207), higher HB trait scores (OR = 1.155), depression (OR = 1.145), and state anger (OR = 1.087). Only 44.54% of the participants were aware that HB is a diagnosed mental disorder and 34.97% knew that it could be treated at traditional Korean medicine (TKM) clinics. Despite the limited awareness, 49% expressed interest in TKM treatment for HB, in which acupuncture, herbal medicine, and mind–body medicine are the preferred modalities. **Conclusions**: This study revealed a high HB prevalence among the MZ generation in the Republic of Korea, which challenges traditional perceptions of this syndrome. These findings highlight the need to reevaluate HB conceptualization and treatment approaches for younger generations. Future research should focus on longitudinal studies, qualitative investigations of the evolving HB cultural context, and the development of targeted awareness and intervention programs to address this significant mental health issue in contemporary Korean society.

## 1. Introduction

Hwa-byung (HB), a Korean culture-bound syndrome, is considered a type of anger syndrome [1,2]. Anger-related syndromes are not unique to Korean culture. For example, ‘intermittent explosive disorder’ and anger attacks in patients with major depressive disorder or panic disorder are recognized in Western psychiatry [3,4,5], while ‘ataques de nervios’ in Latino culture shares similarities with HB [6]. However, the specific cultural context and symptom presentation of HB make it distinct from these other anger-related syndromes [3]. The characteristic of this mental disorder encompasses various psychological and somatic symptoms (heat sensation, pushing-up feeling in the chest, dry mouth, palpitations, and getting angry) resembling ‘fire (hwa in Korean)’ that appear due to suppressed anger [7]. This mental disorder is generally believed to have originated from traditional Korean medicine (TKM), and in 2013, the TKM clinical practice guidelines for HB were developed and disseminated in the Republic of Korea [7]. The Korean Classification of Diseases, based on the International Statistical Classification of Diseases and Related Health Problems, has included TKM-specific disease codes, including HB, since its 6th revision (U22.2) [8]. Therefore, in the Republic of Korea, TKM physicians primarily manage HB treatment.

Conceptually, the disease is considered to be related to perceptions of unfairness and is particularly common among middle-aged Korean women, especially in relation to the Confucian culture of this country [9]. The traditional gender roles defined by Confucianism have pressured Korean women to endure unfair discrimination and the resulting suppression of anger is believed to have created a culture-specific mental disorder called HB [2]. However, mental health problems related to unfair discrimination are not limited to middle-aged women and have recently been emphasized as mental health problems in the general population worldwide [10,11]. Importantly, although HB has traditionally been associated with middle-aged Korean women [9], recent societal changes and evolving gender roles [12] may have altered its presentation and affected the demographics.

The MZ generation encompasses Millennials (individuals born between 1980 and 1994) and Generation Z (individuals born between 1995 and 2005) [13]. This generation has received attention in Korean society due to its unique characteristics that distinguish it from other generations, including an emphasis on personal achievement and work–life balance [14]. Accordingly, this generation values fairness, and awareness of fairness plays an important role in their well-being [15], characterizing them as sensitive to the perception of unfairness. Accordingly, it can be hypothesized that HB caused by the suppression of anger due to unfairness is not rare among this generation in the Republic of Korea. However, the presence of HB, its related factors, and awareness of and attitudes toward HB in the MZ generation have not been sufficiently studied. Considering the importance of mental health issues in Korean adults [16], investigating the effect of HB in the MZ generation may contribute to the development of mental health policies in the Republic of Korea.

Therefore, this study aims to address the gap in the knowledge regarding the prevalence and presentation of HB in younger generations (i.e., the MZ generation). As societal norms and pressures evolve, it is crucial to understand how culture-bound syndromes like HB may manifest differently across generations.

## 2. Materials and Methods

### 2.1. Study Design and Participants

This study employed a cross-sectional design using an anonymous online survey, conducted from 7–12 June 2024, by Macromill Embrain (Embrain Co., Ltd., Seoul, Republic of Korea), a professional survey company. The company sent email invitations containing the survey link to registered panel members who potentially met the inclusion criteria. Macromill Embrain’s panel members are popular research participants in Republic of Korea, frequently involved in large-scale surveys [17,18,19]. Participants were paid a monetary incentive of approximately 0.1 USD per minute for completing this 10 to 15 min online survey [17]. This study was ‘strengthening the reporting of observational studies in epidemiology’ (STROBE)-statement compliant [20]. The inclusion criteria for this study were as follows: (1) Republic of Korea nationality and (2) born between 1980 and 2005. Exclusion criteria included the following: (1) without serious physical illness (such as angina, myocardial infarction, cerebral hemorrhage, cerebral infarction, and cancer) and (2) without serious mental illness (such as depressive disorder, bipolar disorder, and dementia).

### 2.2. Sample Size Calculation

According to the Korean Statistical Information Service, as of 2020, the MZ generation in Republic of Korea included approximately 16.3 million people [13]. Serdar et al.‘s study suggested estimating the sample size based on the population size, recommending a sample of 384 people with a 5% margin of error for populations exceeding 1 million [21]. Accordingly, the current study set its target sample size to 384 people based on their recommendation [21].

### 2.3. Variables

Variables were selected based on existing literature regarding HB and related psychological constructs [2,7,22,23,24]. Demographic variables included age, sex, marital status, region of residence, and subjective health status. The subsections in this section describe the clinical variables.

#### 2.3.1. Hwa-Byung Scale

The HB scale, which consists of 31 questions, was developed by Kwon et al. and evaluates HB traits, which is vulnerability to HB, and HB symptoms, which are the severity of psychological and physical somatic symptoms of HB [23]. A cutoff score of 30 or higher for HB symptoms is considered indicative of the HB presence [23]. Cronbach’s alpha value for the scale used in this study was 0.94.

#### 2.3.2. Emotional Labor

As HB has been conceptualized to be related to suppressed emotional expression [7], participant emotional labor was investigated in this study. Emotional labor was assessed using an instrument developed by Lee. This 14-item scale evaluates employee-focused and job-focused emotional labor [25]. Cronbach’s alpha value for the scale used in this study was 0.89.

#### 2.3.3. Psychological Distress

Psychological distress associated with HB encompasses depression, anxiety, stress, and anger. The Depression, Anxiety, and Stress Scale—21 Items (DASS-21) was used to evaluate depression, anxiety, and stress, consisting of seven questions for each [26]. The State-Trait Anger Expression Inventory (STAXI) was used to assess anger-related symptoms. This tool, developed by Spielberger et al., assesses participant trait anger, state anger, anger-in, anger-out, and anger control using 44 questions [27]. Cronbach’s alpha values for the scales used in this study were as follows: DASS-21 (α = 0.96) and STAXI (α = 0.95).

#### 2.3.4. Perceptions and Attitudes Toward HB

Customized questionnaires were developed to assess participant knowledge, perceptions, and attitudes towards HB. Cronbach’s alpha value for the scale used in this study was 0.75.

### 2.4. Data Analyses

Data analyses were performed using PASW Statistics for Windows (version 18.0; SPSS Inc., Chicago, IL, USA). Descriptive statistics were calculated for all of the variables. Chi-squared and independent *t*-tests were used to compare the characteristics between the HB and non-HB groups. Effect sizes were reported as Cohen’s d for *t*-tests and Cramér’s V for Chi-squared tests. For Cohen’s d, we used the pooled standard deviation method to account for different group sizes. Bivariate logistic regression was employed to identify factors associated with HB presence. Effect sizes are reported as odds ratios (ORs) and 95% confidence intervals (CIs). Statistical significance was set at *p* < 0.05.

### 2.5. Ethical Considerations

This study adhered to the ethical principles outlined in the Declaration of Helsinki. Participants were informed about the study’s purpose, the voluntary nature of participation, and confidentiality of the data. The research protocol was reviewed and approved by the Institutional Review Board of Dong-eui University Korean Medicine Hospital (DH-2024-05, approved on 28 May 2024).

## 3. Results

### 3.1. Baseline Participant Characteristics

A total of 449 participants met the inclusion criteria, including 163 individuals with HB and 286 without HB (Figure 1). The mean age of the participants was 34.67 ± 7.46 years, and the percentage of women was 46.55%. No significant differences were observed in sex, age, generation (M or Z), region, or marital status between the HB and non-HB groups (*p* > 0.05 for all). However, there was a significant difference in subjective health state between the two groups; the HB group was more disadvantaged (*p* < 0.001). HB traits were significantly higher in the HB group (39.07 ± 7.95) compared to the non-HB group (29.48 ± 7.77, *p* < 0.001). All aspects of emotional labor were significantly higher in the HB group (*p* < 0.001 for all subscales). Depression, anxiety, and stress scores of the DASS-21 were significantly higher in the HB group (*p* < 0.001 for all). All STAXI components were significantly higher in the HB group (*p* < 0.001 for most subscales and *p* = 0.005 for anger-control) (Table 1).

### 3.2. Factors Associated with Hwa-Byung Presence

The regression analysis revealed that female sex (odds ratio [OR] = 2.314, 95% CI: 1.123–4.767, *p* = 0.023) was a significant predictor of HB presence compared to male sex. Participants who reported ‘bad or very bad’ subjective health states had a higher OR of HB (OR = 3.207, 95% CI: 1.020–10.084, *p* = 0.046) compared to those reporting ‘good or very good’. In addition, the HB trait score (OR = 1.155, 95% CI: 1.085–1.230, *p* < 0.001), depression (OR = 1.145, 95% CI: 1.018–1.287, *p* = 0.024), and anger states (OR = 1.087, 95% CI: 1.005–1.176, *p* = 0.038) were significant predictors of HB presence (Table 2, Figure 2).

### 3.3. Participant Perceptions and Attitudes Toward Hwa-Byung

Among the respondents, 44.54% (200/449) were aware that HB is a diagnosed mental disorder, with a significantly higher proportion of HB group participants being aware compared to the non-HB group (χ2 = 6.995, *p* < 0.05). Similarly, 34.97% (157/449) knew that HB could be treated at TKM clinics; the HB group had significantly higher awareness (χ2 = 8.307, *p* = 0.005). Regarding HB perceptions, participants generally agreed that it has a significant impact on an individual’s mental health (mean = 4.18 ± 0.82 on a 5-point Likert scale). They also viewed HB as an important public mental health issue comparable to depression (mean = 3.89 ± 0.80). Participants agreed that HB is more common in older adults; the highest agreement for its occurrence was in people in their 40s and 50s (mean = 3.88 ± 0.84), followed by those older than 60 years (mean = 3.73 ± 0.92) (Table 3).

When asked about TKM for potential treatment, 49.00% (220/449) of the participants indicated they would consider TKM treatment for HB, whereas 36.75% (165/449) were unsure. The main reasons for choosing TKM treatment were perceived effectiveness (49.55%, 109/220) and the variety of treatments available (38.64%, 85/220). The preferred TKM treatments for HB included acupuncture (54.55%, 120/220), herbal medicine (52.27%, 115/220), and mind–body medicine (50.91%, 112/220). Participant expectations from TKM treatment primarily focused on symptom improvement and overall health enhancement (57.91%, 260/449), as well as a comfortable and stable atmosphere (49.44%, 222/449). However, concerns about TKM treatment included uncertainty or lack of evidence regarding effectiveness (63.47%, 285/449) and the high cost of services (48.55%, 218/449). Regarding areas for improvement in TKM treatment, participants emphasized the need to accumulate evidence on the effectiveness of TKM services (61.69%, 277/449), reduce costs (45.43%, 204/449), and promote TKM services (42.32%, 190/449) (Table 3).

## 4. Discussion

### 4.1. Main Findings

This study offers crucial insights into the prevalence, associated factors, and perceptions of HB among the MZ generation in the Republic of Korea. HB prevalence in this population was 36.3%, which was substantially higher than expected and challenges the traditional view of HB as a condition that primarily affects middle-aged individuals. Significant predictors of HB included female sex, poor subjective health, higher HB trait and depression scores, and higher state anger. These findings align with previous HB research and extend understanding of the MZ generation. Importantly, the study revealed a significant lack of awareness among participants regarding HB as a diagnosable mental disorder and the availability of treatment in TKM clinics. Only 44.54% of the respondents were aware that HB is a diagnosed mental disorder, and only 34.97% knew that it could be treated at TKM clinics. These findings indicate the presence of HB in this population and that people in this population have significant differences in knowledge and/or awareness of HB.

### 4.2. Implications of the Findings

The high HB prevalence among the MZ generation suggests that this culture-bound syndrome can persist and evolve across generations, adapting to new social contexts while retaining its core features. This has broader implications for cross-cultural psychiatry and the study of how mental health presentations change over time within a society. Among the instruments used in the current survey, question 31 of the HB scale investigates perceptions of unfairness worldwide. Although this was not in our planned analysis of interest, 11.36% of the respondents strongly agreed that the world is unfair, and this proportion was significantly higher in the HB group than in the non-HB group (25.77% vs. 3.15%, *p* < 0.001). Individually perceived discrimination is significantly related to depressive symptoms in young Korean adults [28]; our findings suggest that HB may also be an important mental health problem in this context. The relationship between perceptions of unfairness and HB in the MZ generation reflects the evolving sociocultural context of this syndrome. Although traditional HB is often linked to gender-based discrimination in Confucian societies [9], contemporary manifestations may be more closely tied to broader social inequalities, workplace injustices, and generational conflicts [15,29]. This shift necessitates a re-examination of the social determinants for HB in modern Korean society.

In the current study, the identification of predictive factors, such as female sex, poor subjective health status, high HB trait scores, depression, and state anger, indicated that HB was associated with various psychosocial factors, requiring a multifaceted approach to its management. The higher odds of HB in females suggest that sex-specific stressors or coping mechanisms may still play a crucial role in HB development, even in younger generations. This finding warrants further investigation into sex-specific risk and potential protective factors. For example, the higher prevalence of HB in females in our study warrants deeper consideration in light of recent research on gender-based stereotypes and discrimination. While traditional perspectives often focus on direct discrimination from men, studies in Western contexts have shown that negative stereotypes against women can be perpetuated through complex social dynamics, including internalized biases among women themselves and particularly from older women [30]. This multi-layered nature of gender discrimination might be especially relevant in the Korean context, where traditional gender roles and modern values intersect in the experiences of the MZ generation.

The association between poor subjective health and increased HB risk highlights the intricate relationship between perceived physical well-being and mental health. Poor subjective health is associated with more frequent emotional reactions to stressful events by women and older adults [31]. Additionally, poor subjective health is a predictive factor for the more frequent use of TKM services such as acupuncture [32]. In the current study, the respondents most often chose symptom and overall health improvement expected from TKM treatment for HB. This indicates that promoting overall health and subjective well-being may be key targets for the prevention and/or treatment of HB in TKM clinics. While this study focused on TKM approaches, it is important to note that conventional pharmacotherapy is also used to treat HB in the Republic of Korea. Commonly prescribed medications for HB include selective serotonin reuptake inhibitors and anticonvulsants [1]. However, there is a lack of standardized pharmacological treatment guidelines specifically for HB, and treatment often follows protocols for related conditions such as depression or anxiety disorders [1].

The significant association between HB trait scores and HB presence supports the construct validity of HB as a distinct syndrome and suggests that certain personality traits or cognitive patterns may predispose individuals to HB [23]. Furthermore, according to the current findings, state anger, but not trait anger, was significantly associated with HB presence. This suggests that HB traits are a unique vulnerability to HB that is distinct from trait anger in the STAXI and that HB may be an etiologically complex anger-related syndrome. The link between depression scores and HB indicated potential comorbidities or overlap between these conditions, as previously reported [2]. However, notably, the exclusion criteria for this study included depressive disorder; accordingly, there were no participants who had depressive disorder. These results support the idea that HB is a unique mental disorder related to depression, although it is distinct from depressive disorders. Importantly, this mental disorder is characterized by somatic symptoms; a recent study viewed HB as a functional somatic syndrome [7].

Despite the high HB prevalence in this population, the lack of awareness of HB (only 44.54% recognized it as a diagnosable mental disorder) highlights the need for education and awareness campaigns. Interest in TKM treatment (49% would consider it) suggests that TKM approaches could be valuable in managing the mental health of the MZ generation. However, expectations (symptom improvement, overall health enhancement, and a comfortable atmosphere) and concerns (cost and uncertainty about effectiveness) regarding TKM services indicate areas where TKM should evolve to meet these needs. Thus, the dissemination and promotion of the TKM clinical practice guidelines for HB, which were revised in 2021 [22], can partially resolve the uncertainty regarding the effectiveness of TKM for HB in this generation. Additionally, the results of applying the critical pathway based on this clinical practice guideline suggest the possibility of reducing the costs related to HB treatment at TKM clinics [33]. One of the important findings of this study was that mind–body medicine and psychotherapy were the preferred TKM treatments for HB by the MZ generation. As this survey found, with emotional support, empathy, and thorough explanations expected, TKM doctors need to strengthen evidence-based psychosocial approaches, including mind–body medicine and psychotherapy [34], in their practice.

### 4.3. Suggestions for Further Studies

Based on the findings of the current study, the following suggestions for further research are provided. Longitudinal studies on the long-term course and prognosis of HB in those in the MZ generation are required to understand how HB develops and changes over time. Qualitative research on the relationship between sociocultural factors (perceptions of unfairness and intergenerational conflicts) and HB should be conducted to deepen understanding of the cultural context of HB. Additionally, comparing the differences in HB prevalence and related factors between the MZ generation in other countries with those of the Republic of Korea would be helpful in understanding this culture-related syndrome. To understand the high prevalence of HB, especially in women, future research should investigate how internalized gender stereotypes and multi-generational patterns of discrimination affect HB manifestation in the MZ generation. This could include examining how different sources of gender-based discrimination (from men, female peers, and older women) might differently impact HB development and expression [30]. Such studies could provide valuable insights into the evolving nature of gender-related mental health challenges in contemporary Korean society. Finally, the respondents agreed that HB plays an important role in public mental health, as does depression. Therefore, studies that develop and evaluate the effectiveness of awareness improvement and early intervention programs for HB are necessary to promote mental health in the MZ generation.

### 4.4. Limitations

This study had a few limitations. First, the cross-sectional nature of the study precluded causal inferences about the relationship between HB and its associated factors. Longitudinal studies are needed to establish temporal relationships and causal pathways. Second, the reliance on self-reported measures may have introduced reporting biases, particularly for culturally sensitive topics such as mental health. Future studies could benefit from the incorporation of objective measures or clinician-administered assessments. Third, although the sample size was adequate for the analyses, it may not have been fully representative of the entire MZ generation in the Republic of Korea. Factors such as geographical distribution, socioeconomic status, and educational level should be rigorously controlled in future studies to enhance generalizability. Fourth, although the HB scale used in this study has been validated in Korean populations, there may have been nuances in how the MZ generation interpreted and responded to these measures compared to older generations. Further validation of these instruments, specifically for the MZ generation, is warranted. Fifth, participants who chose to respond to the survey may have had different characteristics from those who did not, which potentially influenced the prevalence estimates and associations observed. Sixth, the lack of a control sample from other generations, such as Generation X or Y, is a limitation of the current study. Given the traditional belief that HB is common in middle-aged women [9], this aspect warrants further investigation. Finally, although not addressed in this study, it would be interesting to explore the potential influence of recent societal changes, evolving gender roles [12], and internalized gender stereotypes on women across generations [35]. Furthermore, given the complex nature of gender-related stereotypes and discrimination, including their internalization and perpetuation within gender groups [30], future studies should incorporate more detailed measures of these factors. Such exploration could provide valuable insights into the changing nature of gender-related stressors in Korean society.

## 5. Conclusions

This study revealed a surprisingly high HB prevalence among the Republic of Korea’s MZ generation, which challenges traditional perceptions of this culture-bound syndrome. The key predictors of HB include female sex, poor subjective health, high HB trait scores, depression, and state anger. Despite the limited awareness of HB as a diagnosable condition treatable with TKM, there was significant interest in TKM treatments among this study population. These findings underscore the need to re-evaluate HB conceptualization and treatment approaches for younger generations, emphasizing the importance of addressing societal inequities in mental health interventions. In addition, this study highlights the ongoing relevance of HB in contemporary Korean society and the potential role of TKM in addressing this cultural syndrome. Future research should focus on longitudinal studies, qualitative investigations of HB’s evolving cultural context, and the development of targeted awareness and intervention programs to better serve the mental health needs of the MZ generation.

## Figures and Tables

**Figure 1 jcm-13-06667-f001:**
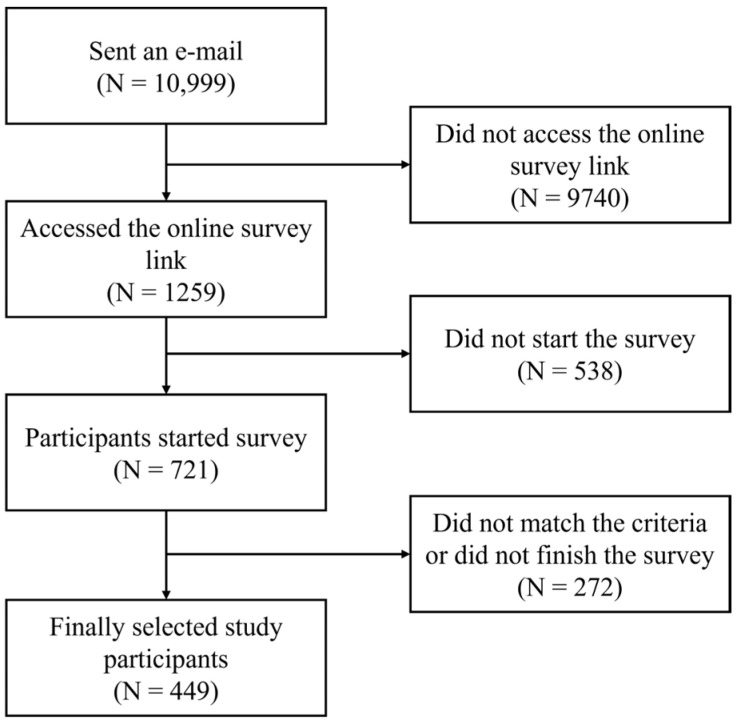
Flow diagram of the selection of the study participants.

**Figure 2 jcm-13-06667-f002:**
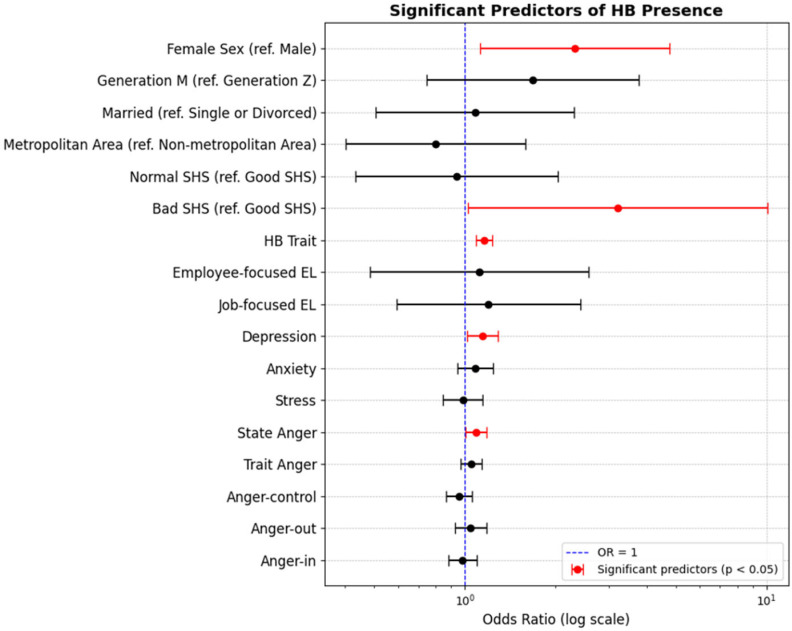
Significant predictors of HB presence. Abbreviations. EL, emotional labor; HB, hwa-byung; OR, odd ratio; SHS, subjective health state.

**Table 1 jcm-13-06667-t001:** Characteristics of the participants.

Variables		Total	Non-HB (n = 286)	HB (n = 163)	Effect Size (Cohen’s d/Cramér’s V)	χ2 or *t*-Value	*p*-Value
Sex	Male	240	156	84	0.028	0.379	0.556
	Female	209	130	79			
Age (year)	19–29 years old	126	82	44	0.073	2.442	0.295
	30–39 years old	148	100	48			
	40–44 years old	175	104	71			
	Raw (years old)	34.67 ± 7.46	34.34 ± 7.53	35.27 ± 7.34	−0.124	−1.276	0.202
Generation	Generation M	261	160	101	0.058	1.546	0.233
	Generation Z	188	126	62			
Region	Metropolitan area	184	117	67	0.002	0.002	1.000
	Non-metropolitan area	265	169	96			
Marriage	Single or divorced	277	181	96	0.043	0.847	0.365
	Married	172	105	67			
Subjective health state	Good or very good	157	124	33	0.320	46.007	0.000 ***
	Normal	221	139	82			
	Bad or very bad	71	23	48			
HB scale	HB Traits	32.96 ± 9.09	29.48 ± 7.77	39.07 ± 7.95	−1.230	−12.483	0.000 ***
	HB Symptoms	25.69 ± 11.61	18.70 ± 7.30	37.94 ± 6.46	−2.764	−27.978	0.000 ***
EL	Employee-focused EL	3.26 ± 0.65	3.09 ± 0.64	3.55 ± 0.56	−0.756	−6.624	0.000 ***
	Superficial acting	3.67 ± 0.76	3.18 ± 0.75	3.69 ± 0.68	−0.708	−6.184	0.000 ***
	Deep acting	3.16 ± 0.75	3.00 ± 0.76	3.41 ± 0.67	−0.570	−4.982	0.000 ***
	Job-focused EL	3.08 ± 0.69	3.09 ± 0.64	3.55 ± 0.56	−0.756	−4.816	0.000 ***
	Frequency of interactions	3.20 ± 0.90	3.05 ± 0.94	3.44 ± 0.76	−0.444	−3.870	0.000 ***
	Duration of interactions	3.20 ± 0.85	3.09 ± 0.86	3.38 ± 0.79	−0.353	−3.101	0.000 ***
	Variety of expression	2.89 ± 0.91	2.75 ± 0.89	3.14 ± 0.90	−0.435	−3.890	0.000 ***
	Total EL	3.16 ± 0.60	3.01 ± 0.60	3.41 ± 0.52	−0.706	−6.266	0.000 ***
DASS-21	Depression	7.02 ± 5.32	4.57 ± 4.05	11.33 ± 4.48	−1.605	−16.369	0.000 ***
	Anxiety	5.69 ± 4.94	3.58 ± 3.58	9.40 ± 4.84	−1.364	−14.533	0.000 ***
	Stress	8.22 ± 4.86	6.05 ± 3.98	12.01 ± 3.82	−1.537	−15.479	0.000 ***
STAXI	State anger	16.66 ± 6.84	13.77 ± 4.94	21.73 ± 6.76	−1.337	−14.307	0.000 ***
	Trait anger	19.68 ± 6.47	17.43 ± 5.30	23.63 ± 6.46	−1.047	−10.974	0.000 ***
	Anger-control	19.37 ± 4.18	18.94 ± 4.25	20.10 ± 3.95	−0.281	−2.853	0.005 **
	Anger-out	14.72 ± 4.63	13.27 ± 3.97	17.26 ± 4.62	−0.935	−9.626	0.000 ***
	Anger-in	17.44 ± 4.98	15.63 ± 4.31	20.61 ± 4.46	−1.140	−11.612	0.000 ***
Any SI	Absence	279	224	55	0.442	87.703	0.000 ***
	Presence	170	62	108			
Much or more SI	Absence	364	265	99	0.392	68.934	0.000 ***
	Presence	85	21	64			
Very much SI	Absence	428	284	144	0.249	27.960	0.000 ***
	Presence	21	2	19			

Abbreviations. DASS-21, Depression, Anxiety, and Stress Scale—21 Items; EL, emotional labor; HB, hwa-byung; SI, suicidal ideation; STAXI, State-Trait Anger Expression Inventory. Note. **, *p* < 0.01; ***, *p* < 0.001.

**Table 2 jcm-13-06667-t002:** Bivariate regression analysis analyzing factors related to the presence of HB.

Variables		OR (95% CIs)	*p*-Value
Sex (ref. male)	Female	2.314 (1.123, 4.767)	0.023 *
Generation (ref. generation Z)	Generation M	1.676 (0.745, 3.770)	0.212
Marriage (ref. single or divorced)	Married	1.079 (0.506, 2.302)	0.845
Region (ref. non-metropolitan area)	Metropolitan area	0.799 (0.402, 1.588)	0.522
Subjective health state (ref. good or very good)	Normal	0.937 (0.432, 2.032)	0.870
	Bad or very bad	3.207 (1.020, 10.084)	0.046 *
HB scale	HB trait	1.155 (1.085, 1.230)	0.000 ***
EL	Employee-focused EL	1.116 (0.484, 2.576)	0.797
	Job-focused EL	1.196 (0.594, 2.409)	0.616
DASS-21	Depression	1.145 (1.018, 1.287)	0.024 *
	Anxiety	1.082 (0.946, 1.237)	0.249
	Stress	0.985 (0.846, 1.146)	0.842
STAXI	State anger	1.087 (1.005, 1.176)	0.038 *
	Trait anger	1.047 (0.965, 1.136)	0.270
	Anger-control	0.957 (0.866, 1.056)	0.380
	Anger-out	1.044 (0.924, 1.179)	0.489
	Anger-in	0.981 (0.882, 1.092)	0.732

Abbreviations. CI, confidence interval; DASS-21, Depression, Anxiety, and Stress Scale—21 Items; EL, emotional labor; HB, hwa-byung; OR, odd ratio; SI, suicidal ideation; STAXI, State-Trait Anger Expression Inventory. Note. *, *p* < 0.05; *** *p* < 0.001.

**Table 3 jcm-13-06667-t003:** Bivariate regression analysis analyzing factors related to the presence of HB.

Variables		Total	Non-HB (n = 286)	HB (n = 163)	Effect Size (Cohen’s d/Cramér’s V)	χ2 or *t*-Value	*p*-Value
Did you know that HB is a diagnosed mental disorder?	I know it.	200	114	86	0.125	6.995	0.010 *
	I do not know it.	249	172	77			
Did you know that HB can be treated at TKM clinics?	I know it.	157	86	71	0.136	8.307	0.005 **
	I do not know it.	292	200	92			
Impression of HB (1–5 point Likert scale, closer to 5 indicating higher level of agreement)	HB occurs in relation to Confucian culture.	3.20 ± 1.00	3.18 ± 1.00	3.23 ± 1.00	−0.050	−0.522	0.602
	HB is a type of anger syndrome.	3.73 ± 0.87	3.71 ± 0.84	3.75 ± 0.93	−0.045	−0.412	0.681
	HB is common in teenagers.	2.94 ± 0.95	2.88 ± 0.93	3.05 ± 0.97	−0.180	−1.851	0.065
	HB is common in people in their 20s and 30s.	3.41 ± 0.89	3.37 ± 0.86	3.49 ± 0.93	−0.135	−1.426	0.155
	HB is common in people in their 40s and 50s.	3.88 ± 0.84	3.87 ± 0.82	3.88 ± 0.87	−0.012	−0.113	0.910
	HB is common in people over 60.	3.73 ± 0.92	3.72 ± 0.89	3.75 ± 0.98	−0.032	−0.378	0.705
	HB is common in women.	3.64 ± 0.93	3.60 ± 0.90	3.72 ± 0.97	−0.129	−1.280	0.201
	HB is common in men.	3.27 ± 0.83	3.24 ± 0.80	3.31 ± 0.88	−0.084	−0.883	0.378
	HB is a mental disorder that requires treatment.	3.62 ± 0.85	3.60 ± 0.78	3.65 ± 0.96	−0.058	−0.588	0.557
	HB has a significant impact on an individual’s mental health.	4.18 ± 0.82	4.22 ± 0.77	4.11 ± 0.90	0.132	1.322	0.187
	HB is a mental health condition as important to public mental health as depression.	3.89 ± 0.80	3.89 ± 0.78	3.88 ± 0.85	0.012	0.181	0.856
	HB is a mental health condition that needs to be managed through the establishment of national health policy.	3.74 ± 0.86	3.72 ± 0.80	3.77 ± 0.95	−0.058	−0.512	0.609
Have you ever received TKM services?	Yes	231	152	79	0.045	0.911	0.377
	No	218	134	84			
If you have HB, would you treat it with TKM treatment?	Yes	220	133	87	0.067	2.015	0.365
	No	64	42	22			
	I do not know.	165	111	54			
Why did you choose yes?	Because of the effectiveness of TKM	109	67	42	NA	NA	NA
	Because of the safety of TKM	64	35	29	NA	NA	NA
	Because of the variety of TKM treatments	85	55	30	NA	NA	NA
	Because of the aversion to conventional medicine	44	32	12	NA	NA	NA
	Other	2	0	2	NA	NA	NA
Preferred TKM treatments	Herbal medicine	115	67	48	NA	NA	NA
	Acupuncture	120	64	56	NA	NA	NA
	Pharmacopuncture	64	42	22	NA	NA	NA
	Moxibustion	65	41	24	NA	NA	NA
	Cupping	48	30	18	NA	NA	NA
	Mind–body medicine	112	75	37	NA	NA	NA
	Psychotherapy	100	60	40	NA	NA	NA
	Other	0	0	0	NA	NA	NA
Why did you choose no?	Because of the high cost of TKM service	16	8	8	NA	NA	NA
	Because of the insufficient or delayed effectiveness of TKM	47	29	18	NA	NA	NA
	Because I am more familiar with conventional medicine.	23	15	8	NA	NA	NA
	Because I have an aversion to TKM.	11	8	3	NA	NA	NA
	This is because there are no TKM clinics near my area of residence.	1	1	0	NA	NA	NA
	Other	0	0	0	NA	NA	NA
What are your expectations from TKM treatment?	Emotional support and empathy from TKM doctors	159	96	63	NA	NA	NA
	Reasonable treatment cost of TKM services	165	97	68	NA	NA	NA
	Symptom improvement and overall health improvement through TKM services	260	165	95	NA	NA	NA
	Comfortable and stable atmosphere	222	149	73	NA	NA	NA
	Full explanation from TKM doctors	127	83	44	NA	NA	NA
	Other	2	1	1	NA	NA	NA
What are your concerns about TKM treatment?	Expensive prices for TKM services	218	140	78	NA	NA	NA
	Delayed therapeutic effect of TKM services	151	81	70	NA	NA	NA
	Uncertainty or lack of evidence regarding the effectiveness of TKM services	285	190	95	NA	NA	NA
	Reluctance or fear of TKM services	67	42	25	NA	NA	NA
	Difficult explanation from TKM doctors	37	20	17	NA	NA	NA
	Inconsistent TKM treatment	133	80	53	NA	NA	NA
	Other	1	0	1	NA	NA	NA
What do you think needs to be improved about TKM treatment?	Cost savings for TKM services	204	131	73	NA	NA	NA
	Improving the treatment effectiveness of TKM services	172	92	80	NA	NA	NA
	Accumulating evidence on the effectiveness of TKM services	277	182	95	NA	NA	NA
	Promotion of TKM services	190	127	63	NA	NA	NA
	Easy explanation from TKM doctors	96	56	40	NA	NA	NA
	Improved consistency of TKM treatment	138	91	47	NA	NA	NA
	Other	1	1	0	NA	NA	NA

Abbreviations. HB, hwa-byung; NA, not applicable; TKM, traditional Korean medicine. Note. *, *p* < 0.05; **, *p* < 0.01.

## Data Availability

The data supporting the findings of this study are available upon request from the authors.

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
