# Peer review of "Hwa-Byung (Anger Syndrome) in the MZ Generation of Republic of Korea: A Survey"

_jcm, 2024, doi:10.3390/jcm13226667_

Round 1
Reviewer 1 Report
Comments and Suggestions for Authors
It is useful to indicate more methodological details. for example, on which platforms was the questionnaire distributed?
How were the interviewees reached? Were they paid to respond? Was the survey advertised?
The research design needs to be more detailed.
The introduction is very good. How, do we move from the generation of middle-aged adult women, hypothetically Generation X, to other generations? What do the data say about the middle-aged generation? What are the social causes of the development of the pathology?
Was a control sample with generation Y done?
Author Response
- Comments from Reviewer 1:
Comment 1: Methodological details
It is useful to indicate more methodological details. for example, on which platforms was the questionnaire distributed? How were the interviewees reached? Were they paid to respond? Was the survey advertised?
Response 1:
Thank you for your valuable feedback. We have added more details about the survey distribution in the Methods section.
“This study employed a cross-sectional design using an anonymous online survey, conducted from June 7 to 12, 2024, by Macromill Embrain (Embrain Co., Ltd., Seoul, Republic of Korea), a professional survey company. The company sent email invitations containing the survey link to registered panel members who potentially met the inclusion criteria. Macromill Embrain's panel members are popular research participants in the Republic of Korea, frequently involved in large-scale surveys [17-19]. Participants were paid a monetary incentive of approximately 0.1 USD per minute for completing this 10- to 15-minute online survey [17].”
(Please refer page 2, red words)
Comment 2: Research design
The research design needs to be more detailed.
Response 2:
Thank you for your comment. We have expanded the description of the research design in the Methods section.
“This study employed a cross-sectional design using an anonymous online survey, conducted from June 7 to 12, 2024, by Macromill Embrain (Embrain Co., Ltd., Seoul, Republic of Korea), a professional survey company.”
(Please refer page 2, red words)
Comment 3: Generational shift in HB
The introduction is very good. How, do we move from the generation of middle-aged adult women, hypothetically Generation X, to other generations? What do the data say about the middle-aged generation? What are the social causes of the development of the pathology?
Response 3:
We thank the reviewer for reading the Introduction part to this manuscript with interest. We have added a paragraph in the Introduction to address the shift from middle-aged women to younger generations.
“However, mental health problems related to unfair discrimination are not limited to middle-aged women and have recently been emphasized as mental health problems in the general population worldwide [10,11]. Importantly, although HB has traditionally been associated with middle-aged Korean women [9], recent societal changes and evolving gender roles [12] may have altered its presentation and affected the demographics.”
(Please refer page 2, red words)
Comment 4: Control sample
Was a control sample with generation Y done?
Response 4:
The survey was not conducted on Generation Y in this study. However, since we have identified the unique epidemiological characteristics of this anger syndrome for Generation MZ in this survey, we plan to conduct the study by expanding the generation. We have added these plans in the Discussion section as follows.
“Finally, the lack of a control sample from other generations, such as Generation X or Y, is a limitation of the current study. Given the traditional belief that HB is common in middle-aged women [9], this aspect warrants further investigation. Although not addressed in this study, it would be interesting to explore the potential influence of recent societal changes, evolving gender roles [12] and internalized gender stereotypes on women across generations [34]. Such exploration could provide valuable insights into the changing nature of gender-related stressors in Korean society.”
(Please refer page 10, red words)

Reviewer 2 Report
Comments and Suggestions for Authors
This study examines Hwa-byung (HB), traditionally associated with middle-aged Korean women, was investigated among the MZ generation in the Republic of Korea to determine its prevalence, associated factors, and perceptions.
Some points for further revision:
Please describe relevant anger syndromes in other countries in the Introduction.
Negative stereotypes against women and discrimination (not only reported by men, but also by other women and especially older women) and their psychological and cognitive health have been supported in other western cultural contexts (e.g. for a relevant recent article on male female humility authors can discuss: doi: 10.1002/brb3.2857). Authors can suggest the future examination of the included variables/factors as a future study to be done in their country on their specific topic in the Future research section and the Limitations of their study (as they are not investigated in their sample).
The rationale of the study is not described in detail and justified.
It is not clear how the sample size was estimated. Please describe in more detail.
Why were these variables entered in this study. Please justify and provide Cronbach's for all included questionnaires.
Effect sizes are not reported for all applied statistical analyses.
Comments on the Quality of English LanguageMinor English language editing.
Author Response
- Comments from Reviewer 2:
Comment 1:
This study examines Hwa-byung (HB), traditionally associated with middle-aged Korean women, was investigated among the MZ generation in the Republic of Korea to determine its prevalence, associated factors, and perceptions.
Response 1:
Thank you for your constructive feedback. We have addressed your comments and suggestions as follows.
Comment 2: Anger syndromes in other countries
Some points for further revision:
Please describe relevant anger syndromes in other countries in the Introduction.
Response 2:
Thank you for your comment. We have added descriptions of relevant anger syndromes in other countries to this revised manuscript, reflecting the reviewer's comments.
“Hwa-byung (HB) is a Korean culture-bound syndrome and has been considered as a type of anger syndrome [1,2]. Anger-related syndromes are not unique to Korean culture. For example, 'intermittent explosive disorder' and anger attacks in patients with major depressive disorder or panic disorder are recognized in Western psychiatry [3-5], while 'ataques de nervios' in Latino culture shares similarities with HB [6]. However, the specific cultural context and symptom presentation of HB make it distinct from these other anger-related syndromes [3]. The characteristic of this mental disorder comprises various psychological and somatic symptoms (heat sensation, pushing-up feeling in the chest, dry mouth, palpitation, and getting angry) similar to “fire (hwa in Korean)” appear due to suppressed anger emotion [7].”
(Please refer page 1, red words)
Comment 3: Negative stereotypes and discrimination
Negative stereotypes against women and discrimination (not only reported by men, but also by other women and especially older women) and their psychological and cognitive health have been supported in other western cultural contexts (e.g. for a relevant recent article on male female humility authors can discuss: doi: 10.1002/brb3.2857). Authors can suggest the future examination of the included variables/factors as a future study to be done in their country on their specific topic in the Future research section and the Limitations of their study (as they are not investigated in their sample).
Response 3:
Thank you for your comment. We believe that addressing negative stereotypes against women and discrimination as recommended by the reviewer would be valuable for our future research. Therefore, we have revised the manuscript as follows.
“Finally, the lack of a control sample from other generations, such as Generation X or Y, is a limitation of the current study. Given the traditional belief that HB is common in middle-aged women [9], this aspect warrants further investigation. Although not addressed in this study, it would be interesting to explore the potential influence of recent societal changes, evolving gender roles [12] and internalized gender stereotypes on women across generations [34]. Such exploration could provide valuable insights into the changing nature of gender-related stressors in Korean society.”
(Please refer page 10, red words)
Comment 4: Rationale of the study
The rationale of the study is not described in detail and justified.
Response 4:
Thank you for your comment. We have expanded the rationale in the Introduction.
“Therefore, this study aims to address the gap in knowledge regarding the prevalence and presentation of HB in younger generations (i.e., MZ generation). As societal norms and pressures evolve, it is crucial to understand how culture-bound syndromes like HB may manifest differently across generations.”
(Please refer page 2, red words)
Comment 5: Sample size estimation
It is not clear how the sample size was estimated. Please describe in more detail.
Response 5:
Thank you for your comment. We have added more details on sample size calculation in the Methods section
“According to the Korean Statistical Information Service, as of 2020, the MZ generation in the Republic of Korea included approximately 16.3 million people [13]. Serdar et al.'s study suggested estimating the sample size based on the population size, recommending a sample of 384 people with a 5% margin of error for populations exceeding 1 million [21]. Accordingly, the current study set its target sample size to 384 people based on their recommendation [21].”
(Please refer page 2, red words)
Comment 6: Variable selection and Cronbach's alpha
Why were these variables entered in this study. Please justify and provide Cronbach's for all included questionnaires.
Response 6:
Thank you for your comment. We have added justification for variable selection and reported Cronbach's alpha values in the Methods section.
“2.3. Variables
Variables were selected based on previous literature on HB and related psychological constructs [2,7,22-24]. Demographic variables included age, sex, marital status, region of residence, and subjective health status. The subsections in this section describe the clinical variables.
2.3.1. Hwa-byung scale
The HB scale, which consists of 31 questions, was developed by Kwon et al. and evaluates HB traits, which is vulnerability to HB, and HB symptoms, which are the severity of psychological and physical somatic symptoms of HB [23]. A cutoff score of 30 or higher for HB symptoms is considered indicative of the HB presence [23]. Cronbach's alpha value for the scale used in this study was 0.94.
2.3.2. Emotional labor
Since HB has been conceptualized as being related to suppressed emotional expression [7], participant emotional labor was investigated in this study. For emotional labor, an instrument developed by Lee was used. This 14-item scale evaluates employee-focused and job-focused emotional labor [25]. Cronbach's alpha value for the scale used in this study was 0.89.
2.3.3. Psychological distress
Psychological distress associated with HB includes depression, anxiety, stress, and anger. The Depression, Anxiety, and Stress Scale-21 (DASS-21) was used to evaluate depression, anxiety, and stress, seven questions for each [26]. The State-Trait Anger Expression Inventory (STAXI) was used to assess anger-related symptoms. This tool, developed by Spielberger et al., assesses participant trait anger, state anger, anger-in, anger-out, and anger control using 44 questions [27]. Cronbach's alpha values for the scales used in this study were as follow: DASS-21 (α = 0.96), and STAXI (α = 0.95).
2.3.4. Perceptions and attitudes toward HB
Customized questionnaires were developed to assess participant knowledge, perceptions, and attitudes towards HB. Cronbach's alpha value for the scale used in this study was 0.75.”
(Please refer page 3, red words)
Comment 7: Effect sizes
Effect sizes are not reported for all applied statistical analyses.
Response 7:
Thank you for your comment. We have added effect sizes (Cohen's d and Cramér's V) for all t-tests and for chi-square tests in the Results section.
“Data analyses were performed using PASW Statistics for Windows (version 18.0; SPSS Inc., Chicago, IL, USA). Descriptive statistics were calculated for all the variables. Chi-square and independent t-tests were used to compare the characteristics between the HB and non-HB groups. Effect sizes were reported as Cohen's d for t-tests and Cramér's V for chi-square tests. For Cohen's d, we used the pooled standard deviation method to account for different group sizes. Bivariate logistic regression was employed to identify factors associated with HB presence. Effect sizes are reported as odds ratios (ORs) and 95% confidence intervals (CIs). Statistical significance was set at p < 0.05.”
(Please refer page 3, red words)
(Please refer Tables 1 and 3, red words)

Reviewer 3 Report
Comments and Suggestions for Authors
I liked this article. The content is well structured, the manuscript is easy to read, and the topic is interesting from a scientific and clinical point of view.
There are several technical comments:
In the "Materials and methods" section: Please add the criteria for excluding participants from the study; do not use abbreviations in the names of subsections.
In the "Results" section: Do not use abbreviations in the names of subsections;
In the “Discussion” section: What pharmacotherapy methods are used in the Republic of Korea to treat anger syndrome?
Author Response
- Comments from Reviewer 3:
Comment 1:
I liked this article. The content is well structured, the manuscript is easy to read, and the topic is interesting from a scientific and clinical point of view.
Response 1:
Thank you for your positive feedback on the structure and content of our manuscript.
Comment 2: Exclusion criteria
There are several technical comments:
In the "Materials and methods" section: Please add the criteria for excluding participants from the study
Response 2:
Thank you for your comment. We have added the exclusion criteria to the Methods section.
“The inclusion criteria for this study were: (1) Republic of Korea nationality; and (2) born between 1980 and 2005. Exclusion criteria included: (1) without serious physical illness (such as angina, myocardial infarction, cerebral hemorrhage, cerebral infarction, and cancer); and (2) without serious mental illness (such as depressive disorder, bipolar disorder, and dementia).”
(Please refer page 2, red words)
Comment 3: Abbreviations in subsection names
In the "Materials and methods" section: do not use abbreviations in the names of subsections.
In the "Results" section: Do not use abbreviations in the names of subsections;
Response 3:
Thank you for your comment. We have removed all abbreviations from subsection names in the Materials and Methods and Results sections.
Comment 4: Pharmacotherapy for anger syndrome
In the “Discussion” section: What pharmacotherapy methods are used in the Republic of Korea to treat anger syndrome?
Response 4:
Thank you for your comment. We have added a brief discussion of pharmacotherapy for HB in the Republic of Korea to the Discussion section
“While this study focused on TKM approaches, it is important to note that conventional pharmacotherapy is also used to treat HB in the Republic of Korea. Commonly prescribed medications for HB include selective serotonin reuptake inhibitors and anticonvulsants [1]. However, there is a lack of standardized pharmacological treatment guidelines specifically for HB, and treatment often follows protocols for related conditions such as depression or anxiety disorders [1].”
(Please refer page 9, red words)

Round 2
Reviewer 2 Report
Comments and Suggestions for Authors
Authors need to expand more in the Discussion on the following:
Negative stereotypes against women and discrimination (not only reported by men, but also by other women and especially older women) and their psychological and cognitive health have been supported in other western cultural contexts (e.g. for a relevant recent article on male female humility authors can discuss: doi: 10.1002/brb3.2857). Authors can suggest the future examination of the included variables/factors as a future study to be done in their country on their specific topic in the Future research section and the Limitations of their study (as they are not investigated in their sample).
Comments on the Quality of English LanguageAn English native speaker should go through the text.
Author Response
Authors’ Response to the Reviewers’ Comments
Journal: Journal of Clinical Medicine
Manuscript number: jcm-3219136
Title: Hwa-byung (anger syndrome) in the MZ Generation of South Korea: a Survey
- Comments from Reviewer 2:
Comment 1:
Authors need to expand more in the Discussion on the following:
*Negative stereotypes against women and discrimination (not only reported by men, but also by other women and especially older women) and their psychological and cognitive health have been supported in other western cultural contexts (e.g. for a relevant recent article on male female humility authors can discuss: doi: 10.1002/brb3.2857). Authors can suggest the future examination of the included variables/factors as a future study to be done in their country on their specific topic in the Future research section and the Limitations of their study (as they are not investigated in their sample).*
Response 1:
Thank you for your thoughtful comment regarding the need to expand our discussion on gender-related stereotypes and discrimination. We agree that this is an important aspect that deserves more attention, and we have made the following modifications to our manuscript.
“For example, the higher prevalence of HB in females in our study warrants deeper consideration in light of recent research on gender-based stereotypes and discrimination. While traditional perspectives often focus on direct discrimination from men, studies in Western contexts have shown that negative stereotypes against women can be perpetuated through complex social dynamics, including internalized biases among women themselves and particularly from older women [30]. This multi-layered nature of gender discrimination might be especially relevant in the Korean context, where traditional gender roles and modern values intersect in the experiences of the MZ generation.”
(Please refer page 9, red words)
“To understand the high prevalence of HB, especially in women, future research should investigate how internalized gender stereotypes and multi-generational patterns of discrimination affect HB manifestation in the MZ generation. This could include examining how different sources of gender-based discrimination (from men, women peers, and older women) might differently impact HB development and expression [30]. Such studies could provide valuable insights into the evolving nature of gender-related mental health challenges in contemporary Korean society.”
(Please refer page 10, red words)
“Furthermore, given the complex nature of gender-related stereotypes and discrimination, including their internalization and perpetuation within gender groups [30], future studies should incorporate more detailed measures of these factors.”
(Please refer page 10, red words)
Comment 2:
An English native speaker should go through the text.
Response 2:
Thank you for the comment. Throughout the manuscript, English expressions have been improved and are highlighted in red.

Round 3
Reviewer 2 Report
Comments and Suggestions for Authors
The changes suggested have been incorporated in the revised text.